



# Technical note: Decomposing a time series into independent trend, seasonal and random components

Dongqin Yin[1,2], Hannah Slatford[1], Michael L. Roderick[1,3]

[1]Research School of Earth Sciences, Australian National University, Canberra, ACT 0200, Australia

[2]Australian Research Council Centre of Excellence for Climate System Science, Canberra, Australia

[3]Australian Research Council Centre of Excellence for Climate Extremes, Canberra, Australia

*Correspondence to*: D. Yin (dongqin.yin@anu.edu.au)



**Abstract:**
Many time series observations in hydrology and climate show large seasonal variations and it has long been
common practice to separate the original data into trend, seasonal and random components. We were interested
in using that decomposition approach as a basis for understanding variability in hydro-climatic time series. For
that purpose, it is desirable that the trend, seasonal and random components are independent so that the variance
of the original time series equals the sum of the variances of the three components. We show that the resulting
decomposition with the trend component traditionally estimated either as a linear trend or a moving average
does not produce components that are independent. Instead we introduce the rarely adopted two-way ANOVA
model into studies of hydro-climatic variability and define the trend as equal to the annual anomaly. This
traditional approach produces a decomposition with three independent components. We then use global land
precipitation data to demonstrate a simple application showing how this decomposition method can be used as a
basis for comparing hydro-climatic variability. We anticipate that the three-part decomposition based on the
two-way ANOVA approach will prove useful for future applications that seek to understand the space-time
dimensions of hydro-climatic variability.
**Keywords:** Time series; Decomposition; Independent component; Climate variability.





**1 Introduction**
Many climatic and hydrologic time series contain large seasonal oscillations and it has long been standard
practice to consider such time series as being composed of three components that include a long-term trend, a
seasonal cycle (or seasonal oscillation) and a random component (Kendall et al., 1983, p. 429; von Storch and
Zwiers, 1999). In practice the trend component is usually removed first using an approach such as (linear) trend
removal (e.g., Kedem and Fokianos, 2002) or sometimes a moving average might be used (e.g., Adhikari and
Agrawal, 2013). Other trend removal techniques are possible (e.g. higher order polynomial, exponential, etc.)
depending on the nature of the time series. Once the trend component has been removed, the mean seasonal
cycle is calculated and the remaining part of the original time series is assigned to the random component. The
details are well known.
Applications of the time series decomposition vary but are usually directed towards analysis and forecasting.
One possible application of the three-part decomposition described above, that is yet to be fully explored in the
climatic and hydrologic sciences is to provide a basis for understanding the variability of a time series. To give
an example, assume we have a monthly precipitation time series that has been decomposed into the above-noted
three components. Once done we can ask how much of the overall variability is due to each of the three parts.
Given that the precipitation time series is the sum of three components, then it follows that the total variance of
the time series is simply the sum of the variances of the three components *plus* three additional terms that
account for the covariances. If the three covariances were all zero, then the partitioning of the total variation
between the components is greatly simplified since the total variance is just the sum of the variances of the three
separate components. A time series decomposition with that property would potentially provide an extremely
useful basis for preparing a climatology of the variability as opposed to a climatology of the mean. For example,
imagine a precipitation time series. By decomposing the original time series into three independent components
we could use a ternary diagram to display, in a single diagram, how the variability is partitioned between those
three components.

The aim of this study is to investigate whether it is possible to identify a time series decomposition approach
that separates a time series into the long-term trend along with seasonal and random components, where the
covariances between the three components are all zero. In other words, the decomposition is such that the three





components are independent. We use monthly precipitation data for various case studies but the underlying
results are equally applicable to other variables (e.g., temperature, runoff, evapotranspiration, etc.). The paper
begins by adopting the standard three-part decomposition described above where we adopt two widely-used
methods to estimate the long-term trend. The first subtracts a linear trend while the second represents the trend
as a moving average. We find that neither of these much-used approaches produces a time series decomposition
with independent components. We then introduce a decomposition method based on the traditional two-way
ANOVA model (e.g., Miller and Kahn, 1962; Sun et al., 2010) where the covariances are all zero. While the
traditional two-way ANOVA model has been widely used in the analysis of scientific experiments it has
received little attention for the analysis of hydro-climatic variability. To demonstrate the application, this
approach is then applied to global land precipitation data to produce maps of the variability with the aim of
showing the potential of the approach.

**2 Precipitation Data**

We use monthly rainfall data from site observations collected by the Australian Bureau of Meteorology
(http://www.bom.gov.au/). We selected three sites to show a variety of different precipitation time series (Fig.
S1). The first is at Darwin Airport (12.42 °S, 130.89 °E, data period: 1941-2017) located in northern Australia.
The precipitation at Darwin Airport has a distinct wet-dry season combined with a long-term upward trend in
precipitation. The results for Darwin Airport are reported in the main text. In the supporting material we show
results at two further sites with very different rainfall characteristics. The second site, Donnybrook (33.57 °S,
115.82 °E, data period: 1906-2017) is located in a winter-dominant precipitation regime in southwest Australia
and shows a long-term decline in precipitation. The final site, Cobar (Lerida) (31.70 °S, 145.70 °E, data period:
1883-1997) is located in the arid centre of New South Wales with precipitation highly variable from year to year
but with no distinct seasonality and no long-term trend.

In a later part of the paper, we use a gridded global precipitation dataset prepared by the Climatic Research Unit
(CRU, TS4.01 database, monthly, 1901-2016, global 0.5° × 0.5°) (Harris et al., 2014), to give an example of
how the two-way ANOVA model can be used to categorize and compare variability.



**3 Statement of the Problem**

We use monthly precipitation time series ($P(t)$) for $q$ years, and separate the time series into components that
describe a long-term trend ($P_a(t)$), monthly means ($P_m(t)$) and a random residual component ($P_r(t)$), such that,
$$P(t) = P_a(t) + P_m(t) + P_r(t) \qquad (1)$$

By the usual variance law, the variance ($\sigma^2$) of $P(t)$ is the sum of variances of each component plus the
covariances (von Storch and Zwiers, 1999),
$$\sigma_P^2 = \sigma_{P_a}^2 + \sigma_{P_m}^2 + \sigma_{P_r}^2 + 2\operatorname{cov}(P_a, P_m) + 2\operatorname{cov}(P_a, P_r) + 2\operatorname{cov}(P_m, P_r) \qquad (2)$$

We test traditional time series decomposition methods and seek a method where the three covariances in Eq. (2)
are all zero.


**4 Evaluating Two Widely-Used Time Series Decomposition Methods**

In this section we use monthly time series for precipitation at Darwin to evaluate whether two widely-used
methods produce decompositions where the individual components are independent (i.e., covariances are zero).
The original data for Darwin cover the period 1941-2017, but we report the decomposition for the shorter period
1942-2016 to account for the loss of data at either end due to the moving average procedure (section 4.2).

**4.1 Time Series Decomposition Using Linear Trend Removal**
On this approach the mean of the time series is first subtracted and a linear regression is fitted to the monthly
anomalies. The resulting regression is then used to calculate the long-term trend component which is
subsequently removed. The monthly means are then calculated and the random component is set equal to the
remainder. The results for Darwin are shown in Fig. 1. (See Figs. S2, S3 for equivalent results at Donnybrook
and Cobar.)

The resulting variance-covariance matrix is shown in Fig. 1e. The overall (temporal) variance of the original
time series is 33716.12 (mm mon$^{-1}$)$^2$. The results show that the variances of the three terms do sum to the total





temporal variance since the least squares estimation is used in the linear regression making the covariances all
sum to zero. However, the individual covariances are not all zero. Actually, when the slope of the linear
regression is not zero (not a constant time series), the covariances between three decomposed components are
also not zero.

**4.2 Time Series Decomposition Using Moving Average Trend Removal**
On this approach the calculation is as before except that a moving average is used to represent the long-term
trend component. In general, one could use a moving average of any period, e.g. months-years-decades. We use
a 24 month moving average but the same general conclusions will hold for other periods. The results for Darwin
are shown in Fig. 2. (See Figs. S4, S5 for equivalent results at Donnybrook and Cobar.)

The resulting variance-covariance matrix is shown in Fig. 2e. Here, the covariances are substantial. For example,
the covariance of the trend and monthly mean components ($\mathrm{cov}(P_\mathrm{a}, P_\mathrm{m}) = 864.00$ (mm mon$^{-1}$)$^2$) is actually larger
than the variance of trend component ($\sigma^2_{P_\mathrm{a}} = 581.34$ (mm mon$^{-1}$)$^2$). The conclusion is that the moving average
method is not suitable for the intended purpose.

**4.3 Summary**
The above evaluation of two widely used traditional methods shows that while the covariances between the
three components were generally (but not always, e.g. covariance value between moving average and monthly
mean components in Fig. 2) small, they were not zero. In the next section, we show a three-part decomposition
method with the desired property that the covariances between the three component are zero.

**5 Introducing a Time Series Decomposition Method based on a Two-way ANOVA Model**

On further investigation we realised that a traditional two-way analysis of variance (ANOVA) model (e.g.,
Miller and Kahn, 1962) which has been widely adopted in designing agricultural experiments (e.g., Clewer and
Scarisbrick, 2001), would meet the criteria we set, i.e., the three components were independent. Briefly, the
temporal mean of the entire (monthly) time series is first subtracted and the anomaly for each year is calculated.





The long-term trend component in each month is calculated by evenly distributing the annual anomaly in each
year to every month in the same year. Once the trend component is extracted from the original time series, the
monthly means are calculated and the random component is set equal to the remainder. It should be noted that in
the traditional two-way ANOVA model, the original time series is actually decomposed into four components,
i.e., long-term mean (constant), net (or centred) annual and monthly components (that have zero means) and the
residual component. In this study, we combine the long-term mean and centred monthly component in the two-
way ANOVA model to produce the monthly means component.

The results for Darwin are shown in Fig. 3. (See Figs. S6, S7 for equivalent results at Donnybrook and Cobar.)
The resulting variance-covariance matrix is shown in Fig. 3e. The covariances are all zero, which demonstrates
that the overall temporal variance (Fig. 3a, $\sigma_P^2 = 33716.12$ (mm mon$^{-1}$)$^2$) is the sum of the variances of the three
independent components. (The same result holds at the Donnybrook and Cobar sites, see Figs. S6 and S7.)  We
further include a mathematical proof (see Appendix) that the covariances are zero in all cases using this
approach. We conclude that a time series decomposition based on the traditional two-way ANOVA model  has
the desired properties.

**6 Variability in Global Precipitation**

We use a global land precipitation database to demonstrate an application of the traditional two-way ANOVA
model decomposition described above.  The data are from the CRU database (monthly, 1901-2016, 0.5° × 0.5°)
where we have calculated the overall temporal variance at each grid-box (Fig. 4a) as well as the percentages of
the total variance due to the annual anomaly (Fig. 4b), monthly (Fig. 4c) and random (Fig. 4d) components.
(The variances for each component are shown in Fig. S8.)

Inspection of Fig. 4a shows that the largest temporal variance of precipitation is generally near the equator. In
tropical Africa and South America, that variation is dominated by the monthly component (Fig. 4c) highlighting
a key point that in these regions the random component of (monthly) precipitation is a relatively small fraction
of the total precipitation. However, that result is not universal throughout the tropics. For example, several
regions throughout South East Asia (e.g., Indonesia, Malaysia) show the opposite pattern with a low fraction of





the total variance due to the monthly (seasonal) component (Fig. 4c) and a correspondingly large fraction due to
the random component. Presumably those parts of South East Asia would also be more drought-prone compared
to tropical Africa and South America. Another key feature is that the fraction of the total variation explained by
the annual (trend) component is small everywhere (Fig. 4b).

To further demonstrate the utility of the approach, we use a ternary diagram to show the fractional partitioning
of the total variance to the three components (Fig. 5). Note that this is only possible because the three
components are independent. In future work we plan a much more comprehensive assessment of hydro-climatic
variability using this approach.

**7 Discussion and Conclusion**

Decomposition of a time series into trend, seasonal and random components has long been used in many
disciplines including studies in hydrology and climate. The emphasis in those studies is often on analysis and
forecasting. However, we were interested in investigating variability and for that application the central attribute
of the chosen decomposition method was whether the covariance between the three components would be zero.
If that were to hold then the total variance would be the sum of the variances of the three components, which
would eliminate the potential complexity arising from the covariance components.

On investigation we found that the two most commonly-used methods for removing the trend (linear and
moving average) will not generally produce components that are independent (Fig. 1, 2). Interestingly, in the
example precipitation time series used here, the moving average approach often produced a covariance between
the trend (24-month moving average) and monthly components that exceeded the variance of trend component
(Figs 2, S4). That approach is clearly not suitable for our intended application. In contrast the linear trend often
produced small covariances with the added feature that the covariance of the trend and monthly components
($\mathrm{cov}(P_a, P_m)$) was the same magnitude but opposite sign from the covariance of annual and random components
($\mathrm{cov}(P_a, P_r)$). This pattern occurs as a design feature of the linear regression method. In particular, the linear
regression produces a trend component ($P_a$) and a remainder ($P_m + P_r$) that are independent by design (i.e.,
$\mathrm{cov}(P_a, P_m + P_r) = 0$). This leads directly to the above-noted cancellation (i.e., $\mathrm{cov}(P_a, P_m) + \mathrm{cov}(P_a, P_r) = 0$), but
the individual covariances are generally not zero.






In contrast the classic two-way ANOVA model separates a time series into trend, monthly and residual
components and was designed to preserve independence among those three components. However, that classic
method has not, to our knowledge, generally adopted to investigate the variability in the hydro-climatic time
series. Our numerical results (Fig. 3, S6, S7) and mathematical proof (Appendix) that the three components are
independent demonstrate the utility of this method in decomposing a time series for studies on variability. One
important point is that the seasonal component (here defined as monthly) repeats over all years of the time series.
Hence caution is needed in applying this approach when it is known that the amplitude of the seasonal
component is changing with time, such as for example, as has been observed for the seasonal cycle of
atmospheric $CO_2$ (Zeng et al., 2014; Piao et al., 2017).

As an application, we applied the two-way ANOVA model to explore the variability in global precipitation. The
temporal variance of precipitation is clearly separated into distinct regimes. In one regime, the total variance is
dominated by the monthly means (seasonal component) while the other regime is dominated by the random
(residual) component. This separation shows good agreement with previous studies based on different
approaches that investigate the predictability of precipitation (Jiang et al., 2016 and 2017). In particular, those
regions with a high predictability of precipitation also have a high fraction of the total variance that is due to the
seasonal component. We expect that a separation of the variance based on this approach will prove useful for
many other applications, especially in studies seeking to understand hydro-climatic variability.

**Data availability**
The monthly rainfall data from site observations can be accessed through the Australian Bureau of Meteorology
(http://www.bom.gov.au/). The global precipitation data is downloaded from the University of East Anglia
Climate Research Unit (CRU): http://data.ceda.ac.uk.

**Author contribution**
D. Yin and M. L. Roderick designed the study and are both responsible for the integrity of the manuscript. D.
Yin and H. Slatford performed the calculations and analysis. D. Yin and M. L. Roderick jointly prepared the
manuscript, and contributed to the interpretation and discussion.




**Competing interests**
The authors declare that there is no conflict of interests.






## Acknowledgements

This research was supported by the Australian Research Council (CE11E0098, CE170100023). The first author

of the paper also acknowledges the support of the National Natural Science Foundation of China (51609122).

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

255

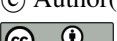



**Appendix: Mathematical Results**

**A.1**      **Independence of the Three Components Using the** Two-way ANOVA Model

Here we show a mathematical derivation of each of the three components based on the two-way ANOVA model

(Section 5 in main text). We use that derivation to demonstrate that the three covariances (Eq. (2) in main text)

all equal zero.

**A.1.1**      **Definition of $P_a(t)$, $P_m(t)$ and $P_r(t)$**

We express the original monthly time series $P(t)$ having dimensions of $q$ years and $p$ ($=12$) months, as a two-

dimensional array,

$$\mathbf{P} = \left[ z_{lk} \right]_{q \times p} \tag{A1}$$

with $l \in [1,q]$ represents order of year, $k \in [1,p]$ represents order of month. Using the matrix subscripts, the

original time series $P(t)$ can be expressed as,

$$
\begin{aligned}
P(t) = [ &\underbrace{z_{11}, \cdots, z_{1k}, \cdots, z_{1p}}_{p\ \text{month}}, && 1^{\text{st}}\ year \\
& \qquad\qquad \vdots \\
&\underbrace{z_{l1}, \cdots, z_{lk}, \cdots, z_{lp}}_{p\ \text{month}}, && l^{\text{th}}\ year \\
& \qquad\qquad \vdots \\
&\underbrace{z_{q1}, \cdots, z_{qk}, \cdots, z_{qp}}_{p\ \text{month}}] && q^{\text{th}}\ year
\end{aligned}
\tag{A2}
$$

We define $u_a(l)$ as the mean in the $l^{\text{th}}$ year,

$$u_a(l) = \frac{\sum\limits_{k=1}^{p} z_{lk}}{p}, \quad l \in [1,q] \tag{A3}$$

and $u_m(k)$ as the mean of the $k^{\text{th}}$ month,

$$u_m(k) = \frac{\sum\limits_{l=1}^{q} z_{lk}}{q}, \quad k \in [1,p] \tag{A4}$$

With $\overline{P(t)}$ the mean of original time series $P(t)$ defined as,





$$\overline{P(t)} = \frac{\sum\limits_{l=1}^{q}\sum\limits_{k=1}^{p} z_{lk}}{q \times p} \tag{A5}$$


we note that $\overline{P(t)}$ is equal to $\overline{u_a(l)}$. To show that, we first calculate $\overline{u_a(l)}$ as,

$$\overline{u_a(l)} = \frac{\sum\limits_{l=1}^{q} u_a(l)}{q} \tag{A6}$$


Combining that with Eq. (A3) and comparing the result with Eq. (A5) we have,

$$\overline{u_a(l)} = \frac{\sum\limits_{l=1}^{q} \dfrac{\sum\limits_{k=1}^{p} z_{lk}}{p}}{q} = \frac{\sum\limits_{l=1}^{q}\sum\limits_{k=1}^{p} z_{lk}}{q \times p} = \overline{P(t)} \tag{A7}$$


Similarly, we calculate $\overline{u_m(k)}$ as,

$$\overline{u_m(k)} = \frac{\sum\limits_{k=1}^{p} u_m(k)}{p} \tag{A8}$$


Combining Eq. (A8) with Eq. (A4) and comparing the result with Eq. (A5) we have,

$$\overline{u_m(k)} = \frac{\sum\limits_{k=1}^{p} \dfrac{\sum\limits_{l=1}^{q} z_{lk}}{q}}{p} = \frac{\sum\limits_{l=1}^{q}\sum\limits_{k=1}^{p} z_{lk}}{q \times p} = \overline{P(t)} \tag{A9}$$



To define the annual component $P_a(t)$ of the decomposition, we first calculate the annual mean in each year, and
using Eq. (A3) we have.

$$P_{\mathrm{annual\,mean}}(l) = \sum\limits_{k=1}^{p} z_{lk} = p \times u_a(l) \tag{A10}$$


Then the anomaly in the $l^{\mathrm{th}}$ year is calculated as,

$$\begin{aligned} \Delta P_{\mathrm{annual\,mean}}(l) &= P_{\mathrm{annual\,mean}}(l) - \overline{P_{\mathrm{annual\,mean}}(l)} \\ &= p \times u_a(l) - \overline{p \times u_a(l)} \\ &= p \times \left( u_a(l) - \overline{u_a(l)} \right) \end{aligned} \tag{A11}$$


Since $\overline{u_a(l)}$ equals $\overline{P(t)}$ (see Eq. (A7)), it follows that Eq. (A11) can be expressed as,

$$\Delta P_{\mathrm{annual\,mean}}(l) = p \times \left( u_a(l) - \overline{P(t)} \right) \tag{A12}$$




We evenly distribute the annual mean anomaly in $l^{\text{th}}$ year (see Eq. (A12)) to all $p$ months in the same year to
define $P_a(t)$ as,

$$P_a(t) = [\underbrace{u_a(1) - \overline{P(t)}, \cdots, u_a(1) - \overline{P(t)}}_{p \text{ month}}, \quad 1^{\text{st}} \; year$$

$$\vdots$$


$$\underbrace{u_a(l) - \overline{P(t)}, \cdots, u_a(l) - \overline{P(t)}}_{p \text{ month}}, \quad l^{\text{th}} \; year \qquad \text{(A13)}$$

$$\vdots$$

$$\underbrace{u_a(q) - \overline{P(t)}, \cdots, u_a(q) - \overline{P(t)}}_{p \text{ month}}] \quad q^{\text{th}} \; year$$


We obtain the monthly mean component $P_m(t)$ by repeating $u_m(k)$ (see Eq. (A4)) for all $q$ years as follows,

$$P_m(t) = [\underbrace{u_m(1), \cdots, u_m(k), \cdots, u_m(p)}_{p \text{ month}}, \quad 1^{\text{st}} \; year$$

$$\vdots$$


$$\underbrace{u_m(1), \cdots, u_m(k), \cdots, u_m(p)}_{p \text{ month}}, \quad l^{\text{th}} \; year \qquad \text{(A14)}$$

$$\vdots$$

$$\underbrace{u_m(1), \cdots, u_m(k), \cdots, u_m(p)}_{p \text{ month}}] \quad q^{\text{th}} \; year$$


With $P(t)$, $P_a(t)$ and $P_m(t)$ now all defined, $P_r(t)$ is the residual component,

$$P_r(t) = P(t) - P_a(t) - P_m(t) \qquad \text{(A15)}$$

and substituting from Eqs. (A2), (A13) and (A14) we have,

$$P_r(t) = [\underbrace{z_{11} - u_a(1) - u_m(1) + \overline{P(t)}, \cdots, z_{1k} - u_a(1) - u_m(k) + \overline{P(t)}, \cdots, z_{1p} - u_a(1) - u_m(p) + \overline{P(t)}}_{p \text{ month}}, \; 1^{\text{st}} \; year$$

$$\vdots$$


$$\underbrace{z_{l1} - u_a(l) - u_m(1) + \overline{P(t)}, \cdots, z_{lk} - u_a(l) - u_m(k) + \overline{P(t)}, \cdots, z_{lp} - u_a(l) - u_m(p) + \overline{P(t)}}_{p \text{ month}}, \; l^{\text{th}} \; year$$

$$\vdots$$

$$\underbrace{z_{q1} - u_a(q) - u_m(1) + \overline{P(t)}, \cdots, z_{qk} - u_a(q) - u_m(k) + \overline{P(t)}, \cdots, z_{qp} - u_a(q) - u_m(p) + \overline{P(t)}}_{p \text{ month}}] \; q^{\text{th}} \; year$$


$$\text{(A16)}$$


**A.1.2   Mean of $P_a(t)$, $P_m(t)$ and $P_r(t)$**





To calculate the covariance, we require the three components (see section A.1.1) and the mean of each
component. We calculate the means in this section and the covariances follow in a later section.

For $P_a(t)$ we take the mean of Eq. (A13),
$$\overline{P_a(t)} = \frac{p \times \sum_{l=1}^{q}\left(u_a(l) - \overline{P(t)}\right)}{q \times p} = \frac{\sum_{l=1}^{q} u_a(l)}{q} - \overline{P(t)} = \overline{u_a(l)} - \overline{P(t)} \qquad (A17)$$

We previously found in Eq. (A7) that $\overline{u_a(l)}$ equals $\overline{P(t)}$, and Eq. (A17) becomes,
$$\overline{P_a(t)} = \overline{u_a(l)} - \overline{P(t)} = 0 \qquad (A18)$$


For $P_m(t)$ we take the mean of Eq. (A14),
$$\overline{P_m(t)} = \frac{q \times \sum_{k=1}^{p} u_m(k)}{q \times p} = \frac{\sum_{k=1}^{p} u_m(k)}{p} = \overline{u_m(k)} \qquad (A19)$$

As $\overline{u_m(k)}$ equals $\overline{P(t)}$ (see Eq. (A9)), then it follows that $\overline{P_m(t)}$ equals $\overline{P(t)}$,
$$\overline{P_m(t)} = \overline{u_m(k)} = \overline{P(t)} \qquad (A20)$$


For $P_r(t)$ we take the mean of Eq. (A15),
$$\overline{P_r(t)} = \overline{P(t)} - \overline{P_a(t)} - \overline{P_m(t)} \qquad (A21)$$

As $\overline{P_a(t)}$ equals zero (see Eq. (A18)) and with $\overline{P_m(t)}$ equal to $\overline{P(t)}$ (see Eq. (A20)), we show that $\overline{P_r(t)}$
equals zero,
$$\overline{P_r(t)} = \overline{P(t)} - \overline{P_a(t)} - \overline{P_m(t)} = \overline{P(t)} - 0 - \overline{P(t)} = 0 \qquad (A22)$$


**A.1.3 Covariance Between the Three Decomposed Components**
Using the above results, we now calculate the (three) covariances (see Eq. (2), main text). We use the sample
covariance but note that the results also hold for the population covariance.

The first (sample) covariance between $P_a(t)$ and $P_m(t)$ is defined by,
$$\mathrm{cov}\left(P_a(t), P_m(t)\right) = \frac{\sum_{l=1}^{q}\sum_{k=1}^{p}\left(\left(P_a(t) - \overline{P_a(t)}\right)\left(P_m(t) - \overline{P_m(t)}\right)\right)}{q \times p - 1} \qquad (A23)$$





Combining Eqs. (A13) and (A18) for the first bracketed term along with Eqs. (A14) and (A20) for the second
bracketed term in the numerator we can rewrite Eqs. (A23) as,

$$
\mathrm{cov}\left(P_a(t), P_m(t)\right) = \frac{\sum_{l=1}^{q}\sum_{k=1}^{p}\left(\left(u_a(l) - \overline{P(t)} - \overline{u_a(l) - \overline{P(t)}} - 0\right)\left(u_m(k) - \overline{u_m(k)}\right)\right)}{q \times p - 1}
$$


$$
= \frac{\sum_{l=1}^{q}\sum_{k=1}^{p}\left(\left(u_a(l) - \overline{u_a(l)}\right)\left(u_m(k) - \overline{u_m(k)}\right)\right)}{q \times p - 1}
$$

(A24)

For the first part of the numerator $\left(u_a(l) - \overline{u_a(l)}\right)$ in Eq. (A24), there is no change for the summation over
index $k$ and therefore this term can be set as a constant for the second summation, and we have,

$$
\mathrm{cov}\left(P_a(t), P_m(t)\right) = \frac{\sum_{l=1}^{q}\left(\left(u_a(l) - \overline{u_a(l)}\right)\sum_{k=1}^{p}\left(u_m(k) - \overline{u_m(k)}\right)\right)}{q \times p - 1}
$$


(A25)

Now that the summation has been separated into two terms, we note that the second summation in Eq. (A25) is
zero. To show that, we note that the mean is the sum divided by number of samples (see Eq. (A8)), and the
second summation can be written as,

$$
\sum_{k=1}^{p}\left(u_m(k) - \overline{u_m(k)}\right) = p \times \left(\frac{\sum_{k=1}^{p}u_m(k)}{p} - \overline{u_m(k)}\right)
$$


$$
= p \times \left(\overline{u_m(k)} - \overline{u_m(k)}\right)
$$

(A26)

$$
= 0
$$

It follows that the covariance between $P_a(t)$ and $P_m(t)$ must be zero,
$$
\mathrm{cov}\left(P_a(t), P_m(t)\right) = 0
$$

(A27)


The (sample) covariance between $P_a(t)$ and $P_r(t)$ is defined by,
$$
\mathrm{cov}\left(P_a(t), P_r(t)\right) = \frac{\sum_{l=1}^{q}\sum_{k=1}^{p}\left(\left(P_a(t) - \overline{P_a(t)}\right)\left(P_r(t) - \overline{P_r(t)}\right)\right)}{q \times p - 1}
$$

(A28)





Then we calculate the covariance between $P_a(t)$ and $P_r(t)$ by introducing definitions of these two terms in Eq.
(A13) and (A16), with the results from Eq. (A18) and (A22), i.e., $\overline{P_a(t)}$ and $\overline{P_r(t)}$ both equal zero. With those
substitutions, we have,
$$\mathrm{cov}\left(P_a(t), P_r(t)\right) = \frac{\sum_{l=1}^{q}\sum_{k=1}^{p}\left(\left(u_a(l) - \overline{P(t)} - 0\right)\left(z_{lk} - u_a(l) - u_m(k) + \overline{P(t)} - 0\right)\right)}{q \times p - 1} \quad (A29)$$

As before, for the first part of the numerator $\left(u_a(l) - \overline{P(t)}\right)$ in Eq. (A29), there is no change for the
summation over index $k$ and therefore this term can be set as a constant for the second summation, and we have,
$$\mathrm{cov}\left(P_a(t), P_r(t)\right) = \frac{\sum_{l=1}^{q}\left(\left(u_a(l) - \overline{P(t)}\right)\sum_{k=1}^{p}\left(z_{lk} - u_a(l) - u_m(k) + \overline{P(t)}\right)\right)}{q \times p - 1} \quad (A30)$$

Again the second summation in the numerator equals zero. To show that, we re-express the second summation
in Eq. (A30) as,
$$\sum_{k=1}^{p}\left(z_{lk} - u_a(l) - u_m(k) + \overline{P(t)}\right) = \sum_{k=1}^{p}z_{lk} - p \times u_a(l) - \sum_{k=1}^{p}u_m(k) + p \times \overline{P(t)} \quad (A31)$$

and after further rearrangement we have,
$$\sum_{k=1}^{p}\left(z_{lk} - u_a(l) - u_m(k) + \overline{P(t)}\right) = p \times \left(\frac{\sum_{k=1}^{p}z_{lk}}{p} - u_a(l)\right) - p \times \left(\frac{\sum_{k=1}^{p}u_m(k)}{p} - \overline{P(t)}\right) \quad (A32)$$

The first term inside the first set of brackets equals $u_a(l)$ (see Eq. (A3)), and the first term inside the second set
of brackets equals $\overline{P(t)}$ (see Eq. (A9)). With those substitutions, Eq. (A32) becomes,
$$\sum_{k=1}^{p}\left(z_{lk} - u_a(l) - u_m(k) + \overline{P(t)}\right) = p \times 0 - p \times 0 = 0 \quad (A33)$$

It follows that the covariance between $P_a(t)$ and $P_r(t)$ is zero,
$$\mathrm{cov}\left(P_a(t), P_r(t)\right) = 0 \quad (A34)$$






Finally, we calculate the covariance between $P_m(t)$ and $P_r(t)$,
$$\mathrm{cov}\left(P_{\mathrm{m}}(t), P_{\mathrm{r}}(t)\right) = \frac{\sum_{l=1}^{q}\sum_{k=1}^{p}\left(\left(P_{\mathrm{m}}(t) - \overline{P_{\mathrm{m}}(t)}\right)\left(P_{\mathrm{r}}(t) - \overline{P_{\mathrm{r}}(t)}\right)\right)}{q \times p - 1} \tag{A35}$$

With previous definitions of $P_m(t)$ and $P_r(t)$ (see Eq. (A14) and (A16)), and results from Eq. (A20) and (A22),
i.e., $\overline{P_m(t)}$ equals $\overline{P(t)}$ and $\overline{P_r(t)}$ equals zero, we have,
$$\mathrm{cov}\left(P_{\mathrm{m}}(t), P_{\mathrm{r}}(t)\right) = \frac{\sum_{l=1}^{q}\sum_{k=1}^{p}\left(\left(u_m(k) - \overline{P(t)}\right)\left(z_{lk} - u_a(l) - u_m(k) + \overline{P(t)} - 0\right)\right)}{q \times p - 1} \tag{A36}$$

As before, for the first part of the numerator $\left(u_m(k) - \overline{P(t)}\right)$ in Eq. (A36), there is no change for the
summation over index $l$ and therefore this term can be set as a constant for the first summation, and we have,
$$\mathrm{cov}\left(P_{\mathrm{m}}(t), P_{\mathrm{r}}(t)\right) = \frac{\sum_{k=1}^{p}\left(\left(u_m(k) - \overline{P(t)}\right)\sum_{l=1}^{q}\left(z_{lk} - u_a(l) - u_m(k) + \overline{P(t)}\right)\right)}{q \times p - 1} \tag{A37}$$

Again the second summation of the numerator equals zero. To show that, we re-express the second summation
in Eq. (A37) as,
$$\sum_{l=1}^{q}\left(z_{lk} - u_a(l) - u_m(k) + \overline{P(t)}\right) = \sum_{l=1}^{q} z_{lk} - q \times u_m(k) - \sum_{l=1}^{q} u_a(l) + q \times \overline{P(t)} \tag{A38}$$

and after further rearrangement we have,
$$\sum_{l=1}^{q}\left(z_{lk} - u_a(l) - u_m(k) + \overline{P(t)}\right) = q \times \left(\frac{\sum_{l=1}^{q} z_{lk}}{q} - u_m(k)\right) - q \times \left(\frac{\sum_{l=1}^{q} u_a(l)}{q} - \overline{P(t)}\right) \tag{A39}$$

The first term inside the first set of brackets equals $u_m(k)$ (see Eq. (A4)), and the first term inside the second
set of brackets equals $\overline{P(t)}$ (see Eq. (A7)). With those substitutions, Eq. (A39) becomes,
$$\sum_{l=1}^{q}\left(z_{lk} - u_a(l) - u_m(k) + \overline{P(t)}\right) = q \times 0 - q \times 0 = 0 \tag{A40}$$

It follows that the covariance between $P_m(t)$ and $P_r(t)$ is zero,

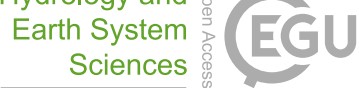

$$\mathrm{cov}\left(P_{\mathrm{m}}(t), P_{\mathrm{r}}(t)\right) = 0 \tag{A41}$$


In summary, all three covariance terms, $\mathrm{cov}\left(P_{a}(t), P_{m}(t)\right)$ (see Eq. (A27)), $\mathrm{cov}\left(P_{a}(t), P_{r}(t)\right)$ (see Eq.
(A34)) and $\mathrm{cov}\left(P_{\mathrm{m}}(t), P_{\mathrm{r}}(t)\right)$ (see Eq. (A41)) are shown to be equal to zero.

In this study we have used 12 (monthly) periods per year. The same results would hold for other time
periods, such as 4 seasons or 365 days per year.

**A.2     Variance of the Random Component**
While undertaking the mathematical analysis we noticed another interesting result, that the variance of the
random component $\sigma^{2}_{P_{r}(t)}$ can be expressed as the sum of the variances calculated for each of the individual
months. We did not use this result, but we anticipate that it will be useful in further applications. For that
purpose, we show the derivation here.

**A.2.1     Sample Variance**

The sample variance of residual component $P_{\mathrm{r}}(t)$ is defined by,
$$\sigma^{2}_{P_{\mathrm{r}}(t)} = \frac{\sum_{k=1}^{p}\sum_{l=1}^{q}\left(P_{\mathrm{r}}(t) - \overline{P_{\mathrm{r}}(t)}\right)^{2}}{q \times p - 1} \tag{A42}$$

With previous definitions of $P_{\mathrm{r}}(t)$ (see Eq. (A16)), and results from Eq. (A22), i.e., $\overline{P_{r}(t)}$ equals zero, we have,
$$\sigma^{2}_{P_{\mathrm{r}}(t)} = \frac{\sum_{k=1}^{p}\sum_{l=1}^{q}\left(z_{kl} - u_{a}(l) - u_{m}(k) + \overline{P(t)}\right)^{2}}{q \times p - 1} \tag{A43}$$


We extract the residual component for each $k^{\mathrm{th}}$ month and define it as $P_{\mathrm{r},k}(t)$,
$$P_{\mathrm{r},k}(t) = \underbrace{[z_{k1} - u_{a}(1) - u_{m}(k) + \overline{P(t)}, \cdots, z_{kl} - u_{a}(l) - u_{m}(k) + \overline{P(t)}, \cdots, z_{kq} - u_{a}(q) - u_{m}(k) + \overline{P(t)}]}_{q \text{ year}}$$

$$\tag{A44}$$





To calculate the sample variance of $P_{r,k}(t)$, we require its mean. For $P_{r,k}(t)$ we take the mean of Eq. (A44),

$$\overline{P_{r,k}(t)} = \frac{\sum_{l=1}^{q}\left(z_{kl} - u_a(l) - u_m(k) + \overline{P(t)}\right)}{q}$$

409                                                                                                            (A45)

$$= \frac{\sum_{l=1}^{q} z_{kl}}{q} - \frac{\sum_{l=1}^{q} u_a(l)}{q} - u_m(k) + \overline{P(t)}$$

The first term in Eq. (A45) equals $u_m(k)$ (see Eq. (A4)) and the second term equals $\overline{P(t)}$ (see Eq. (A7)).
With those substitutions, we have,

412         $$\overline{P_{r,k}(t)} = u_m(k) - \overline{P(t)} - u_m(k) + \overline{P(t)} = 0$$                    (A46)


Based on the above results, we now calculate the sample variance of $P_{r,k}(t)$,

415         $$\sigma^2_{P_{r,k}(t)} = \frac{\sum_{l=1}^{q}\left(P_{r,k}(t) - \overline{P_{r,k}(t)}\right)^2}{q-1}$$    (A47)

With definitions of $P_{r,k}(t)$ (see Eq. (A44)), and results from Eq. (A46), i.e., $\overline{P_{r,k}(t)}$ equals zero, we have,

417         $$\sigma^2_{P_{r,k}(t)} = \frac{\sum_{l=1}^{q}\left(z_{kl} - u_a(l) - u_m(k) + \overline{P(t)} - 0\right)^2}{q-1}$$    (A48)

To show the relation between $\sigma^2_{P_r(t)}$ and $\sigma^2_{P_{r,k}(t)}$ we calculate the sum of $\sigma^2_{P_{r,k}(t)}$ ,

$$\sum_{k=1}^{p}\sigma^2_{P_{r,k}(t)} = \sum_{k=1}^{p}\frac{\sum_{l=1}^{q}\left(z_{kl} - u_a(l) - u_m(k) + \overline{P(t)}\right)^2}{q-1}$$

419                                                                                                            (A49)

$$= \frac{\sum_{k=1}^{p}\sum_{l=1}^{q}\left(z_{kl} - u_a(l) - u_m(k) + \overline{P(t)}\right)^2}{q-1}$$


Comparing Eq. (A49) with Eq. (A43), we have,



$$\sigma^2_{P_r(t)} = \frac{\sum_{k=1}^{p}\sum_{l=1}^{q}\left(z_{kl} - u_a(l) - u_m(k) + \overline{P(t)}\right)^2}{q \times p - 1}$$ (A50)

$$= \frac{q-1}{q \times p - 1} \times \sum_{k=1}^{p}\sigma^2_{P_{r,k}(t)}$$

The result in Eq. (A50) indicates that sample variance of the random component $\sigma^2_{P_r(t)}$ can be expressed as the
sum of the sample variances calculated for each of the individual months.

**A.2.2   Population Variance**
The population variance of residual component $P_r(t)$ is defined by,
$$\widehat{\sigma}^2_{P_r(t)} = \frac{\sum_{k=1}^{p}\sum_{l=1}^{q}\left(P_r(t) - \overline{P_r(t)}\right)^2}{q \times p}$$ (A51)

With definitions of $P_r(t)$ in Eq. (A16), and results from Eq. (A22), i.e., $\overline{P_r(t)}$ equals zero, we have,
$$\widehat{\sigma}^2_{P_r(t)} = \frac{\sum_{k=1}^{p}\sum_{l=1}^{q}\left(z_{kl} - u_a(l) - u_m(k) + \overline{P(t)}\right)^2}{q \times p}$$ (A52)


We now calculate the population variance of $P_{r,k}(t)$,
$$\widehat{\sigma}^2_{P_{r,k}(t)} = \frac{\sum_{l=1}^{q}\left(P_{r,k}(t) - \overline{P_{r,k}(t)}\right)^2}{q}$$ (A53)

With definitions of $P_{r,k}(t)$ (see Eq. (A44)), and $\overline{P_{r,k}(t)}$ equals zero (see Eq. (A46)), we have,
$$\widehat{\sigma}^2_{P_{r,k}(t)} = \frac{\sum_{l=1}^{q}\left(z_{kl} - u_a(l) - u_m(k) + \overline{P(t)} - 0\right)^2}{q}$$ (A54)

To show the relation between $\widehat{\sigma}^2_{P_r(t)}$ and $\widehat{\sigma}^2_{P_{r,k}(t)}$ we calculate the sum of $\widehat{\sigma}^2_{P_{r,k}(t)}$,





$$\sum_{k=1}^{p} \hat{\sigma}^2_{P_{\mathrm{r},k}(t)} = \sum_{k=1}^{p} \frac{\sum_{l=1}^{q} \left( z_{kl} - u_a(l) - u_m(k) + \overline{P(t)} \right)^2}{q}$$


(A55)

$$= \frac{\sum_{k=1}^{p} \sum_{l=1}^{q} \left( z_{kl} - u_a(l) - u_m(k) + \overline{P(t)} \right)^2}{q}$$


Comparing Eq. (A52) with Eq. (A55), we have,

$$\hat{\sigma}^2_{P_{\mathrm{r}}(t)} = \frac{\sum_{k=1}^{p} \sum_{l=1}^{q} \left( z_{kl} - u_a(l) - u_m(k) + \overline{P(t)} \right)^2}{q \times p}$$


(A56)

$$= \frac{q}{q \times p} \times \sum_{k=1}^{p} \hat{\sigma}^2_{P_{\mathrm{r},k}(t)}$$

$$= \frac{1}{p} \times \sum_{k=1}^{p} \hat{\sigma}^2_{P_{\mathrm{r},k}(t)}$$

The result in Eq. (A56) indicates that population variance of the random component $\hat{\sigma}^2_{P_{\mathrm{r}}(t)}$ is the mean of the
population variances calculated for each of the individual months.










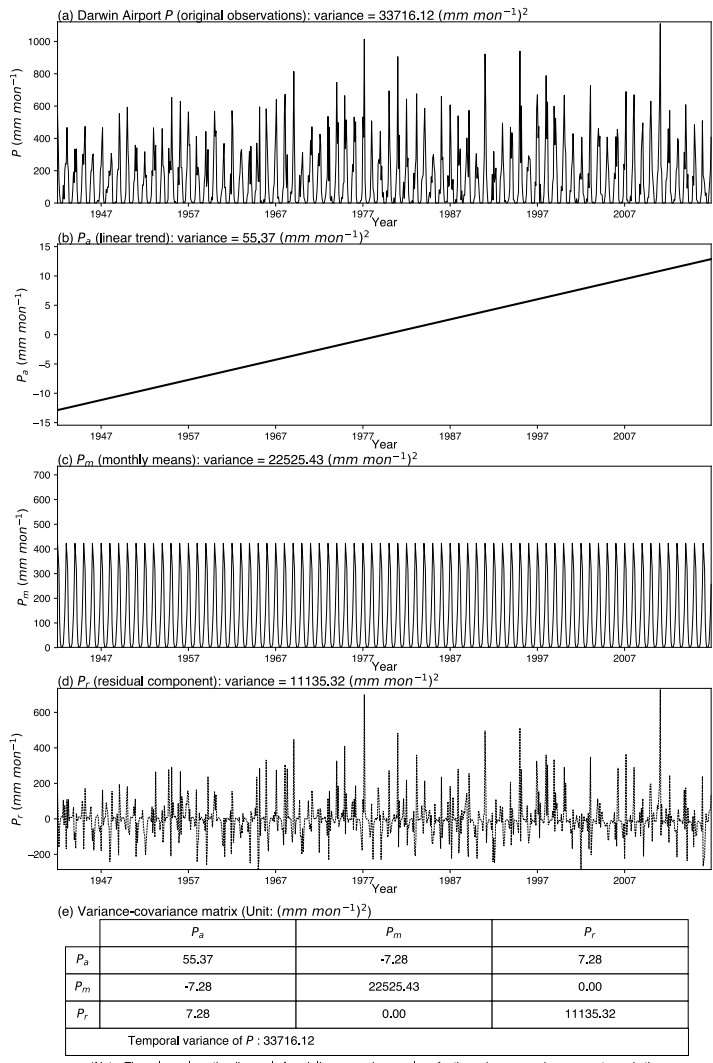


Figure 1. Decomposition of monthly precipitation time series at Darwin (1942-2016) using linear trend removal. Panels
show the (a) original observations ($P$), (b) linear trend ($P_a$), (c) monthly means ($P_m$), (d) residual random component ($P_r$) and
the (e) variance-covariance matrix for the three components ($P_a$, $P_m$ and $P_r$).




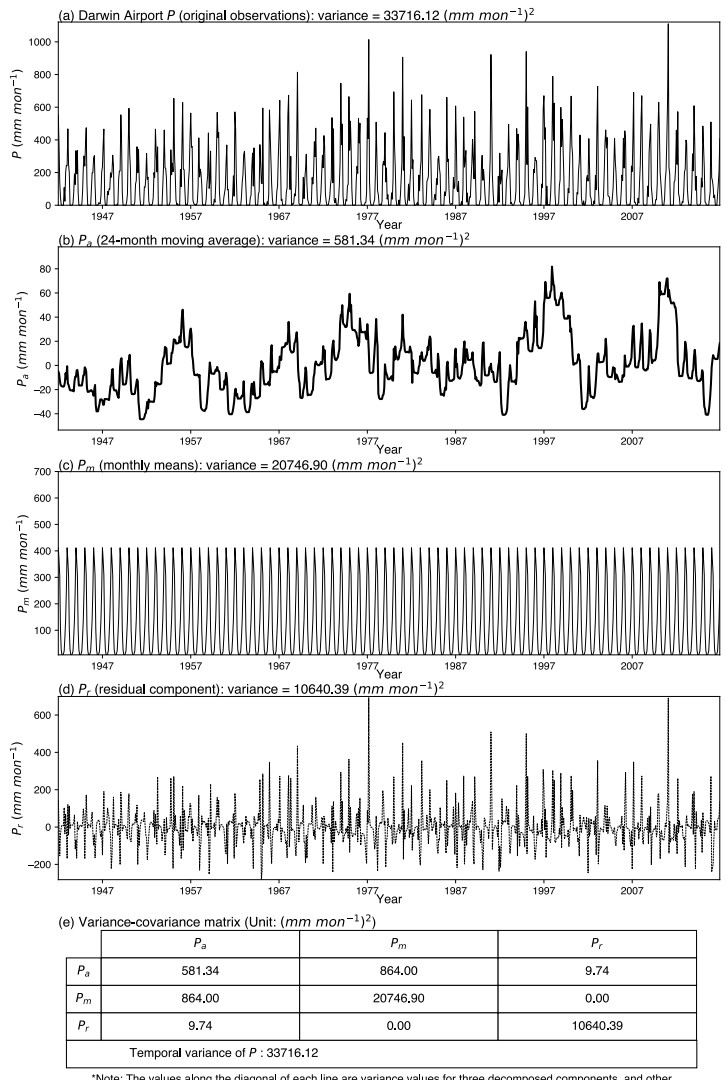


Figure 2. Decomposition of monthly precipitation time series at Darwin (1942-2016) using 24-month moving average trend
removal. Panels show the (a) original observations ($P$), (b) 24-month moving average trend ($P_a$), (c) monthly means ($P_m$), (d)
residual random component ($P_r$) and the (e) variance-covariance matrix for the three components ($P_a$, $P_m$ and $P_r$).




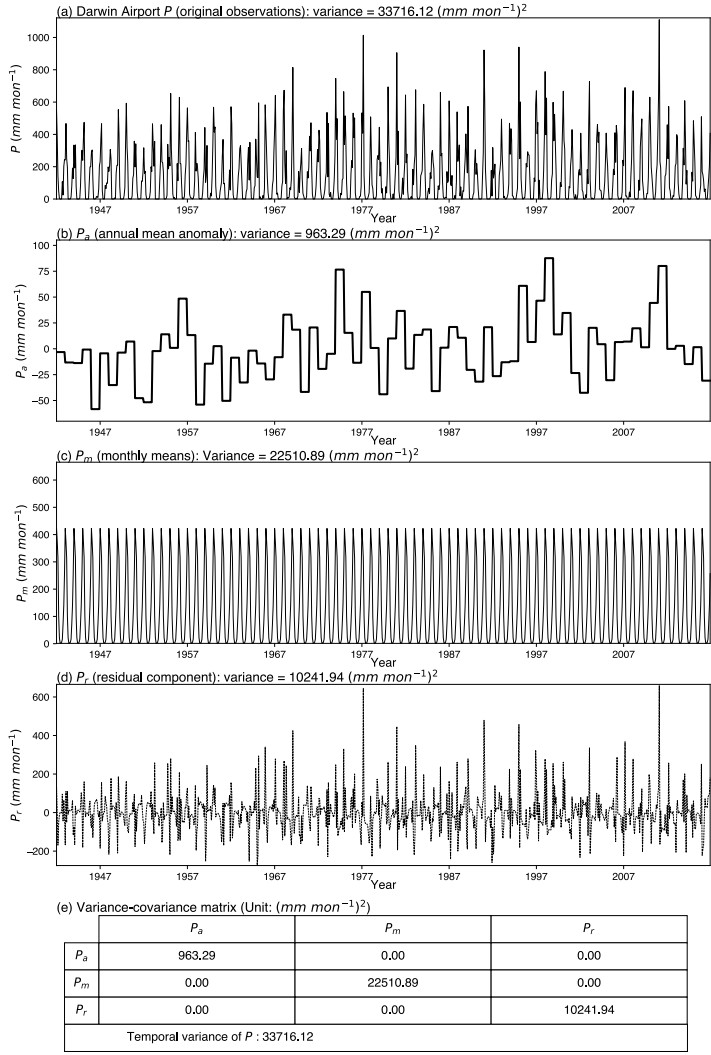


Figure 3. Decomposition of monthly precipitation time series at Darwin (1942-2016) using the two-way ANOVA model.
Panels show the (a) original observations ($P$), (b) annual anomaly ($P_a$), (c) monthly means ($P_m$), (d) residual random
component ($P_r$) and the (e) variance-covariance matrix for the three components ($P_a$, $P_m$ and $P_r$).





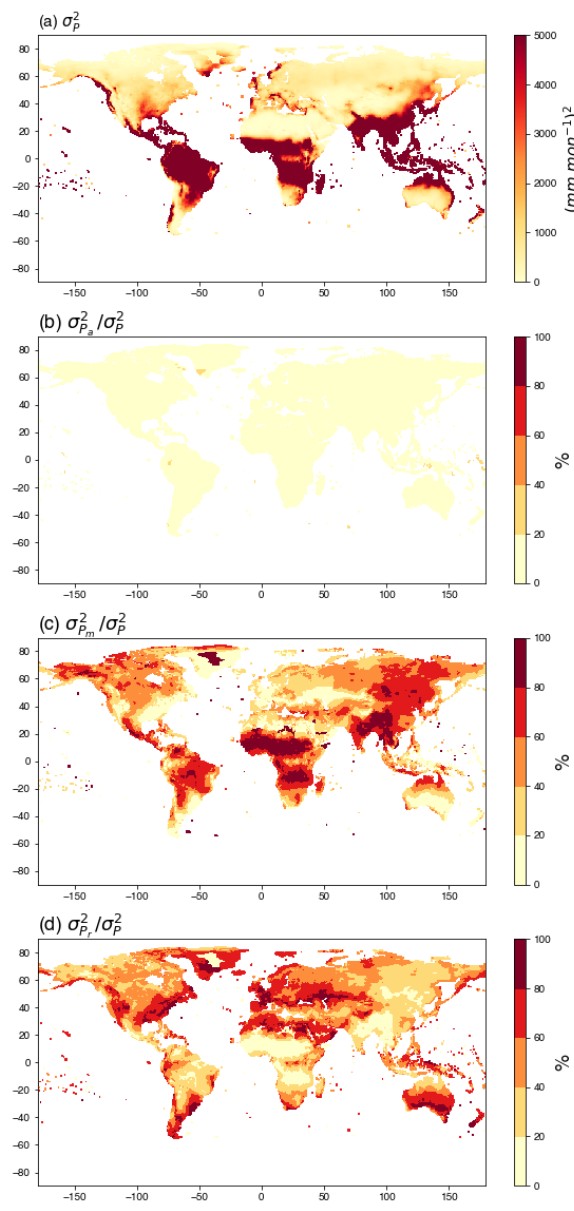


Figure 4. Variability of global land precipitation based on the CRU database (1901-2016) using the two-way ANOVA model.

(a) Temporal variance ($\sigma_P^2$) and fractional contributions due to (b) annual ($\sigma_{P_a}^2 / \sigma_P^2$), (c) monthly ($\sigma_{P_m}^2 / \sigma_P^2$) and (d) random

($\sigma_{P_r}^2 / \sigma_P^2$) variations.




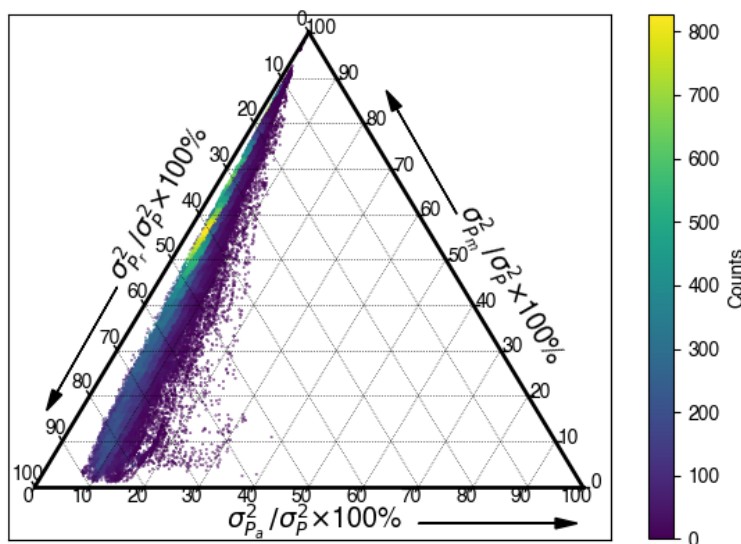


Figure 5. Ternary diagram showing decomposition of the temporal variance ($\sigma_P^2$) into the three independent components
using the two-way ANOVA model. Axes show fractional variance in the annual anomaly ($\sigma_{P_a}^2$), monthly means ($\sigma_{P_m}^2$) and
residual ($\sigma_{P_r}^2$) components.