# Peer review of "Technical note: Decomposing a time series into independent trend, seasonal and random components"

_Hydrology and Earth System Sciences, 2018_

## Referee Comment (RC1) · Anonymous Referee #1 · 30 Dec 2018

The authors of this paper seek a method that decomposes a time series of monthly means into three components, one of which might contain a trend component, the second representative of the annual cycle, and the third representing the remaining stochastic month-to-month variability, with the decomposition being performed in such a way that the pair-wise covariance between the three components is zero. They propose an analysis of variance that achieves this objective. Competing decomposition methods are briefly assessed and found not to produce components with zero pairwise covariance (i.e., are uncorrelated), although the author's calculations suggest that the covariances obtained using these methods are small in most cases in the examples considered.

First, it is not obvious that the three methods considered correspond to three identical models for the month-to-month variation in precipitation – and indeed, it seems that the models cannot be the same. A challenge from the outset is that it is not clear what parts of model (1) are stochastic, and which are fixed, although for all three it seems that the component $P_m(t)$ represents the annual cycle, and thus has the property $P_m(t) = P_m(t+12)$. I would conclude then that $P_m(t)$ is deterministic rather than stochastic, so the notion of variance and covariance (between random variables) doesn't quite seem to fit the bill.

For the first variant of the model, discussed in 4.1, the component $P_a(t)$ is take to be a linear trend. Since the entire observational record is considered, it seems that for this method at least, $P_a(t)$ is deterministic, and thus again, the notion of variance and covariance between random variables does not apply. In this case, what is being fitted is a variant of model (1) with a deterministic linear trend, a fixed annual cycle, and residual stochastic variability. There are probably a number of ways this could be fitted other than by estimating components sequentially – and perhaps an ANOVA formulation is one of those better ways, but this should be judged in terms of the relative efficiency of parameter estimates obtained via different methods as opposed to whether the apparent covariance between components is zero.

The second variant of the model, discussed in 4.2, uses a crude low pass filter to obtain $P_a(t)$. This is rather different from a trend formulation, because the filter will pass both any deterministic change in level over time (assuming that such changes only occur on long time scales) and stochastic variability at time scales that the filter allows to pass. Note that the moving average filter does not cut off smoothly with frequency (it has messy "side lobes" that leak high frequency variance), resulting in contamination of the low frequency component by higher frequency "noise". Thus, this variant of model (1) has in mind trend plus stochastic low-frequency variability as one component, the annual cycle as a second component, and stochastic high-frequency variability as a third component. Variability and covariance of the first and third terms make sense, in so much as deterministic trend is not present in the first term. Clearly this is a different animal from that considered in the first variant of the model. Both the first term is different, and the nature of the variability that is retained in the third term is different.

The third variant of the model, discussed in Section 5, apparently uses a 2-factor ANOVA model to decompose a timeseries of monthly means into an annual effect (with a different level for each year), a month effect (with a different level for each of the 12 months of the year, thus representing the annual cycle), and a residual component. The interpretation of this type of model requires consideration of whether year and month effects are fixed or random. In this case, I would assume that year effects are random, and month effects are fixed, since they are common to all years. The partitioning of variability in an ANOVA analysis is done in such a way that, under the assumption of the Gaussian distribution and iid residual variability, the three variance components that result are statistically independent. This is all standard stuff, and I'm not sure that the long appendix is required to make essentially this point. From a climatological perspective, the interpretation of this variant of model (1) is not very different from that of the second variant of the model considered in section 4.2. The annual component presumably has deterministic trend and low frequency stochastic components, the annual cycle is deterministic, and the residual has higher frequency stochastic variability.

Again, differences between these methods, and the underlying variants of model (1) that are implicit in these methods, should be considered in terms of the different objectives of the methods (a different variant of model (1) implies a somewhat different objective), and whether one method produces better estimates (from a statistical perspective) of model parameters and properties rather than rather arbitrarily focusing on a single aspect, the covariance of component estimates.

One final note concerning a statement that appears on line 90 of the manuscript – zero covariance is synonymous with independence ONLY if the monthly time series values are Gaussian (i.e., normally) distributed. It seems very evident from the figures in the supplement that this is certainly not the case at Cobar (Lerida), e.g., see Figs S3d and S5d. While less pronounced, the similar figures for the other two locations also show some evidence of skewed residuals, and hence a departure from the Gaussian assumption.

---

## Referee Comment (RC2) · Patidar (Referee) · 11 Jan 2019

This paper aims to investigate the potential of time-series decomposition approach for exploring variability in hydro-climatic time series and the possibility of generating independent components (i.e. a zero covariance between the three components). To achieve this, the paper analysed the potential of using a two-way ANOVA based decomposition approach in comparison to the conventional approach based on linear trend removal and using moving average based trend removal. I think paper presented an interesting application of the time series decomposition approach in pattern recognition and the contribution made by the paper could be of interest to the wider hydrologic

community. However, the paper needs significant improvements, specifically, the author should consider addressing some major issues specified below to improve the overall quality of their paper:

1. Literature review: A range of time-series decomposition approaches has been proposed recently and these approaches have been applied to a wide area of application. Literature review/background information presented in the paper is considerably limited. I think the author of the paper should have discussed key work done in the development of time-series decomposition approaches, potential areas of application, and specifically in the context of hydrological applications.

2. Novelty: This paper will also benefit if the author further contextualises the overall purpose of the paper to the state-of-the-art to clearly state/identify the novel contribution of this work.

3. Methodology: I think that the description provided for the methodological approaches is not detailed enough to access their overall appropriateness for the present work. For example in section 4.1 First paragraph – "On this approach ... equal to the remainder", it is not clear if author subtracted mean of long time-series from the monthly mean or annual mean of long time-series is used to estimate monthly anomaly. How they estimated seasonal and random components, just stating that "random component is set equal to the remainder" could be misleading, until they clarify how monthly means have been used in estimating seasonal components (are they using any differencing approach here for estimation of seasonal components or they are just using monthly mean). I think the author should provide mathematical expressions to clarify their procedure. Similarly, methodologies presented in section 4.2 and 4.3 can be updated to improve the technical representation of the paper. Finally, I think methodologies are not critically discussed to justify their appropriateness in this paper. For example: Section 4.2 line 110-111 "In general, ... other period". No explanation/justification is provided for selection of 24 month period for the proposed method and what could be the potential impacts/advantage/disadvantage of using any other larger/smaller time

periods. Section 5 starts with the sentence "On further investigation . . ." and made a concluding statement on two-way analysis of variance (ANOVA) model facilitating three independent components (line 129). It is not clear if the author conducted any sort of pre-investigation before reaching to these conclusions, what other approaches if they investigated, what leads them to the selection of ANOVA model proposed here, any theoretical/technical aspect of ANOVA model that could have resulted in delivering three independent components. As said before the mathematical representation of procedure should be provided to enhance the overall quality of the paper.

4. Results: Results and outcome of the various decomposition approaches are minimally discussed; moreover discussion provided is not technical and is mainly based on general observations only. For example: In Section 4.1 Why the individual covariances are not all zero (line 104). Similarly, in Section 4.2 line 117, the author concluded "the moving average . . . intended purpose". They could explain technical reasons for why they observed larger covariance for Pa and Pm than variance for Pa. Further, what could be possible impact of using moving average; what are the strengths and what are the weaknesses?

5. Other: Section 6 – adequate details should be provided on the various characteristic of global land precipitation database, which are used to demonstrate the application of ANOVA based decomposition model (e.g. what is the structure of the database, what is the temporal/spatial resolution, etc.). Section 7 – (as said before) Results are not critically discussed; I would like to see some critical discussion on the theoretical/technical aspect of the outcomes. It would be good if the author focuses a bit on the novelty aspect of their work and also on what is the contribution of their work.

6. Appendix A.1: I think this section is useful and the author presented a clear mathematical proof to demonstrate all the three components are independent. Some general points – Eq. A13 and A14 - Why there is the same expression for each year in the right-hand side of the equation. The author can provide a brief explanation for this to support non-mathematical experts. Eq. A19 - Why is not equal to 0. Eq. A24 – I think

author miss to put bracket for term ((U_a (l)- (P(t)) ÌĚ ) ) ÌĚ

7. Appendix A.2: I think this analysis is not relevant to this paper or please provide a strong explanation to support, in particular, what could be potential significance/application of possibility of expressing the variance of the random component as the sum of the variance calculated for individual months. How do these findings are related to the autocorrelation properties of the random component, is it 0 for all lags, which implies that the remainder is a purely stochastic process, i.e. white noise process. What could be the significance of having the random component as a white noise process?

Please also note the supplement to this comment:
https://www.hydrol-earth-syst-sci-discuss.net/hess-2018-601/hess-2018-601-RC2-supplement.pdf

---

## Referee Comment (RC3) · Patidar (Referee) · 18 Jan 2019

Dear Author

Thanks for your reply. I appreciate your response and initiatives to improve the overall quality of the paper. You responded well to most of the comments. For Eq A24 In numerator within two summation sign I think you should have ((Ua(I)-P(t) - (Ua(I)-P(t)) - 0) instead of ((Ua(I)-P(t) - (Ua(I)-P(t)) - 0) to allow cancellation of term P(t). Please note that I did not included bars here due to the format of text allowed. Please clarify if I am misunderstanding you and please feel free to contact me or leave a comment if you need further clarification on any of my comments. I looking forward to reading the

updated manuscript.

Sandhya
* * *

---

## Author Comment (AC1) · 18 Jan 2019

**In the following we use R1C1 (etc) to refer to comment 1 (C1) by referee 1 (R1).**

R1C1: The authors of this paper seek a method that decomposes a time series of monthly means into three components, one of which might contain a trend component, the second representative of the annual cycle, and the third representing the remaining stochastic month-to-month variability, with the decomposition being performed in such a way that the pair-wise covariance between the three components is zero. They propose an analysis of variance that achieves this objective. Competing decomposition methods are briefly assessed and found not to produce components with zero pairwise covariance (i.e., are uncorrelated), although the author's calculations suggest that the covariances obtained using these methods are small in most cases in the examples considered.

Response: We appreciate the anonymous review for the evaluation and comment on the manuscript.

R1C2: First, it is not obvious that the three methods considered correspond to three identical models for the month-to-month variation in precipitation – and indeed, it seems that the models cannot be the same. A challenge from the outset is that it is not clear what parts of model (1) are stochastic, and which are fixed, although for all three it seems that the component $P_m(t)$ represents the annual cycle, and thus has the property $P_m(t)= P_m(t+12)$. I would conclude then that $P_m(t)$ is deterministic rather than stochastic, so the notion of variance and covariance (between random variables) doesn't quite seem to fit the bill.

For the first variant of the model, discussed in 4.1, the component Pa(t) is take to be a linear trend. Since the entire observational record is considered, it seems that for this method at least, Pa(t) is deterministic, and thus again, the notion of variance and covariance between random variables does not apply. In this case, what is being fitted is a variant of model (1) with a deterministic linear trend, a fixed annual cycle, and residual stochastic variability. There are probably a number of ways this could be fitted other than by estimating components sequentially – and perhaps an ANOVA formulation is one of those better ways, but this should be judged in terms of the relative efficiency of parameter estimates obtained via different methods as opposed to whether the apparent covariance between components is zero.

The second variant of the model, discussed in 4.2, uses a crude low pass filter to obtain Pa(t). This is rather different from a trend formulation, because the filter will pass both any deterministic change in level over time (assuming that such changes only occur on long time scales) and stochastic variability at time scales that the filter allows to pass. Note that the moving average filter does not cut off smoothly with frequency (it has messy "side lobes" that leak high frequency variance), resulting in contamination of the low frequency component by higher frequency "noise". Thus, this variant of model (1) has in mind trend plus stochastic low frequency variability as one component, the annual cycle as a second component, and stochastic high-frequency variability as a third component. Variability and covariance of the first and third terms make sense, in so much as deterministic trend is not present in the first term. Clearly this is a different animal from that considered in the first variant of the model.

Both the first term is different, and the nature of the variability that is retained in the third term is different.

The third variant of the model, discussed in Section 5, apparently uses a 2-factor ANOVA model to decompose a time series of monthly means into an annual effect (with a different level for each year), a month effect (with a different level for each of the 12 months of the year, thus representing the annual cycle), and a residual component. The interpretation of this type of model requires consideration of whether year and month effects are fixed or random. In this case, I would assume that year effects are random, and month effects are fixed, since they are common to all years. The partitioning of variability in an ANOVA analysis is done in such a way that, under the assumption of the Gaussian distribution and iid residual variability, the three variance components that result are statistically independent. This is all standard stuff, and I'm not sure that the long appendix is required to make essentially this point. From a climatological perspective, the interpretation of this variant of model (1) is not very different from that of the second variant of the model considered in section 4.2. The annual component presumably has deterministic trend and low frequency stochastic components, the annual cycle is deterministic, and the residual has higher frequency stochastic variability.

Again, differences between these methods, and the underlying variants of model (1) that are implicit in these methods, should be considered in terms of the different objectives of the methods (a different variant of model (1) implies a somewhat different objective), and whether one method produces better estimates (from a statistical perspective) of model parameters and properties rather than rather arbitrarily focusing on a single aspect, the covariance of component estimates.

Response: We agree with R1 about the importance of things like distinguishing deterministic and stochastic variables for the three methods, etc. On reading the comment in totality we realize that there is a misunderstanding of our objective. This misunderstanding has arisen because we have not clearly communicated the objective. The original objective of this study is to seek a decomposition method that could be applied to understand the variability of a time series in the hydrologic and climatic sciences. In that context, we do not aim to propose a new decomposition method or develop the existing decomposition methods. Instead, we aim to compare the most commonly used decomposition methods in hydrologic and climatic sciences. For example, partitioning variability in precipitation into the various components to explore how variability of the source (left hand-side of Eq. 2) is partitioned into the various time scales, i.e., sinks (right hand-side of Eq. 2).

In terms of the decomposition of variability, the problem here is the existence of covariances between decomposed components, since the physical meanings of these covariances are difficult to explain. The key to simplify the problem is to have zero covariances between the three decomposed components. Under that circumstance, we have done the comparisons between decomposition methods in this study mainly based on covariance assessment. If the zero covariances is true, the total variance of the time series could be easily attributed to the variances of decomposed components. That will substantially simplify analysis of variability

in the water cycle, and the example application (Fig. 4-5) was intended to provide the necessary context.

However, based on the high-level comments of R1, it is clear that we have not explained our motivation very well. In response, we propose a number of major changes including,

1) change the title of the manuscript to 'Technical note: Decomposing a time series into uncorrelated annual, monthly and residual components for application in variability studies', to more clearly describe the contents,

2) completely rewrite the Introduction to more clearly explain the objective of this study as described above.

We believe that the proposed changes will greatly improve this 'technical note'.

R1C3: One final note concerning a statement that appears on line 90 of the manuscript – zero covariance is synonymous with independence ONLY if the monthly time series values are Gaussian (i.e., normally) distributed. It seems very evident from the figures in the supplement that this is certainly not the case at Cobar (Lerida), e.g., see Figs S3d and S5d. While less pronounced, the similar figures for the other two locations also show some evidence of skewed residuals, and hence a departure from the Gaussian assumption.

Response: Yes, we agree that we have incorrectly used the term 'independent' according to the strict definition. Based on the comment, we have carefully checked the difference between definitions of several terms, e.g., 'independent', 'orthogonal' and 'uncorrelated', which have subtle mathematical distinctions and usually make people confused. Strictly speaking, in this study we try to compare covariances between the decomposed components from different methods, and this is the definition of 'uncorrelated'.

In response, we propose to change the term 'independent' to 'uncorrelated' throughout. We also propose to change the term 'random' in the title and elsewhere in the manuscript to the more accurate term 'residual'. Finally, we propose to change the title of the manuscript to 'Technical note: Decomposing a time series into uncorrelated annual, monthly and residual components for application in variability studies', which we believe follows the important suggestions made by R1.

---

## Author Comment (AC2) · 18 Jan 2019

**In the following we use R2C1 (etc) to refer to comment 1 (C1) by reviewer 2 (R2).**

**Dr. S. Patidar**

R2C1: This paper aims to investigate the potential of time-series decomposition approach for exploring variability in hydro-climatic time series and the possibility of generating independent components (i.e. a zero covariance between the three components). To achieve this, the paper analysed the potential of using a two-way ANOVA based decomposition approach in comparison to the conventional approach based on linear trend removal and using moving average based trend removal. I think paper presented an interesting application of the time series decomposition approach in pattern recognition and the contribution made by the paper could be of interest to the wider hydrologic. community.

Response: We thank Dr. Patidar for the encouraging evaluation and comments on the manuscript.

However, the paper needs significant improvements, specifically, the author should consider addressing some major issues specified below to improve the overall quality of their paper:

R2C2: Literature review: A range of time-series decomposition approaches has been proposed recently and these approaches have been applied to a wide area of application. Literature review/background information presented in the paper is considerably limited. I think the author of the paper should have discussed key work done in the development of time-series decomposition approaches, potential areas of application, and specifically in the context of hydrological applications.

Response: Yes, we agree the literature review for the development and application of key time-series decomposition methods in this study is limited. The reason for that is this was submitted as a 'technical note' and not a full paper. In response to the comment, we propose to add a brief review (approximately one or two paragraphs) describing the decomposition methods mentioned by R2.

R2C3: Novelty: This paper will also benefit if the author further contextualises the overall purpose of the paper to the state-of-the-art to clearly state/identify the novel contribution of this work.

Response: Briefly, our goal is to use the variance equation to partition variability in precipitation (source: left hand-side of Eq. 2) into the various components (sinks: right hand-side of Eq. 2). Inspection of the variance equation (Eq. 2) reveals three covariance terms. Our ultimate goal is to have a decomposition scheme where the three covariance terms are zero. That will simplify analysis of variability of the water cycle substantially. The application (Fig. 4-5) was intended to provide the necessary context. Based on the comment by R2, we have not done that well. We will therefore try to restate more clearly the objective of this study based on the brief literature review in the revised manuscript, and believe that the proposed changes will make the contribution more clear.

R2C4: Methodology: I think that the description provided for the methodological approaches is not detailed enough to access their overall appropriateness for the present work. For example in section 4.1 First paragraph – "On this approach…equal to the remainder", it is not clear if author subtracted mean of long time-series from the monthly mean or annual mean of long time-series is used to estimate monthly anomaly. How they estimated seasonal and random components, just stating that "random component is set equal to the remainder" could be misleading, until they clarify how monthly means have been used in estimating seasonal components (are they using any differencing approach here for estimation of seasonal components or they are just using monthly mean). I think the author should provide mathematical expressions to clarify their procedure. Similarly, methodologies presented in section 4.2 and 4.3 can be updated to improve the technical representation of the paper. Finally, I think methodologies are not critically discussed to justify their appropriateness in this paper. For example: Section 4.2 line 110-111 "In general, … other period". No explanation/justification is provided for selection of 24 month period for the proposed method and what could be the potential impacts/advantage/disadvantage of using any other larger/smaller time periods. Section 5 starts with the sentence "On further investigation…" and made a concluding statement on two-way analysis of variance (ANOVA) model facilitating three independent components (line 129). It is not clear if the author conducted any sort of pre-investigation before reaching to these conclusions, what other approaches if they investigated, what leads them to the selection of ANOVA model proposed here, any theoretical/technical aspect of ANOVA model that could have resulted in delivering three independent components. As said before the mathematical representation of procedure should be provided to enhance the overall quality of the paper.

Response: As suggested, we will add more details of the calculations for three methods in Section 4.1-4.3 in the revised manuscript. We had already included the theoretical/mathematical derivations for the ANOVA model in the Appendix A1.

R2C5: Results: Results and outcome of the various decomposition approaches are minimally discussed; moreover discussion provided is not technical and is mainly based on general observations only. For example: In Section 4.1 Why the individual covariances are not all zero (line 104). Similarly, in Section 4.2 line 117, the author concluded "the moving average… intended purpose". They could explain technical reasons for why they observed larger covariance for Pa and Pm than variance for Pa. Further, what could be possible impact of using moving average; what are the strengths and what are the weaknesses?

Response:  In the manuscript the result sections (Sections 4-6) simply present the outcome, and all the discussion of results is put in the 'Discussion and Conclusion' (Section 7). Therefore, the reason why the individual covariances are not all zero is discussed in Section 7 (please see lines 186-189).

We believe this research is most suitably presented as a 'technical note' and the HESS guidelines are 'a few pages only'. The manuscript as submitted would be about four pages in the printed journal (excluding the mathematical appendices), and it is therefore difficult to make the work longer while still meeting the guidelines for a 'technical note'.

R2C6: Other: Section 6 – adequate details should be provided on the various characteristic of global land precipitation database, which are used to demonstrate the application of ANOVA based decomposition model (e.g. what is the structure of the database, what is the temporal/spatial resolution, etc.). Section 7 – (as said before) Results are not critically discussed; I would like to see some critical discussion on the theoretical/technical aspect of the outcomes. It would be good if the author focuses a bit on the novelty aspect of their work and also on what is the contribution of their work.

Response: The details of the global land precipitation database are included in 'Precipitation Data' (please see lines 71-73). In terms of the statement of the contribution and more discussion of results, please see the response to R2C3 and R2C5 respectively.

R2C7: Appendix A.1: I think this section is useful and the author presented a clear mathematical proof to demonstrate all the three components are independent. Some general points – Eq. A13 and A14 - Why there is the same expression for each year in the right-hand side of the equation. The author can provide a brief explanation for this to support non-mathematical experts. Eq. A19 - Why is not equal to 0. Eq. A24 – I think author miss to put bracket for term $((U\_a (l)- (P(t)) Ì ˇE ) ) Ì ˇE$

Response: Eq. A13 represents the annual mean anomaly time series (Pa), which shares the same annual mean anomaly for each month in the same year but can be different between years. Eq. A14 represents the seasonal cycle time series (Pm), which shares the same mean monthly values in all years. The reason why there is the same expression for each year is that the mean monthly component repeats for all years. With respect to Eq. A19, we did not understand the question because that is overall mean for the seasonal cycle (Pm), which is generally greater than zero. For Eq. A24, we are afraid that we did not understand the comment and did not identify a missing bracket or other problem with Eq. A24.

R2C8: Appendix A.2: I think this analysis is not relevant to this paper or please provide a strong explanation to support, in particular, what could be potential significance/application of possibility of expressing the variance of the random component as the sum of the variance calculated for individual months. How do these findings are related to the autocorrelation properties of the random component, is it 0 for all lags, which implies that the remainder is a purely stochastic process, i.e. white noise process. What could be the significance of having the random component as a white noise process?

Response: We originally included Appendix A2 because we found the results interesting but on reflection we accept the point made by the reviewer that it does not directly contribute to the objective of the 'technical note'. With that, we propose to delete it in the revised manuscript.

---

## Referee Comment (RC4) · Patidar (Referee) · 22 Jan 2019

Thanks, You pointed it out right. I hope you must understand algebra and sign rules for operating the bracket. To cancel P(t) you need to enclose the third and fourth term within the brackets.

---

## Author Comment (AC3) · 22 Jan 2019

**In the following we use R2C1 (etc) to refer to comment 1 (C1) by reviewer 2 (R2).**

**Dr. S. Patidar**

R2C9: Dear Author

Thanks for your reply. I appreciate your response and initiatives to improve the overall quality of the paper. You responded well to most of the comments. For Eq A24 In numerator within two summation sign I think you should have ((Ua(I)-P(t) - (Ua(I)-P(t)) - 0) instead of ((Ua(I)-P(t) - (Ua(I)-P(t)) - 0) to allow cancellation of term P(t). Please note that I did not included bars here due to the format of text allowed. Please clarify if I am misunderstanding you and please feel free to contact me or leave a comment if you need further clarification on any of my comments. I looking forward to reading the updated manuscript.

Response: Eq. A24 as submitted (cut and paste from the .pdf) is,

$$\text{cov}\left(P_a(t), P_m(t)\right) = \frac{\sum_{l=1}^{q}\sum_{k=1}^{p}\left(\left(u_a(l) - \overline{P(t)} - \overline{u_a(l) - \overline{P(t)}} - 0\right)\left(u_m(k) - \overline{u_m(k)}\right)\right)}{q \times p - 1}$$

$$= \frac{\sum_{l=1}^{q}\sum_{k=1}^{p}\left(\left(u_a(l) - \overline{u_a(l)}\right)\left(u_m(k) - \overline{u_m(k)}\right)\right)}{q \times p - 1} \tag{A24}$$

We are not sure but we think you are referring to the third and fourth terms inside the first bracket. The $\overline{P(t)}$ will cancel since $\overline{P(t)}$ equals $\overline{\overline{P(t)}}$. Maybe the best way to avoid confusion is to add brackets around the third and fourth terms like,

$$\text{cov}\left(P_a(t), P_m(t)\right) = \frac{\sum_{l=1}^{q}\sum_{k=1}^{p}\left(\left(u_a(l) - \overline{P(t)} - \left(\overline{u_a(l) - \overline{P(t)}}\right) - 0\right)\left(u_m(k) - \overline{u_m(k)}\right)\right)}{q \times p - 1}$$

$$= \frac{\sum_{l=1}^{q}\sum_{k=1}^{p}\left(\left(u_a(l) - \overline{u_a(l)}\right)\left(u_m(k) - \overline{u_m(k)}\right)\right)}{q \times p - 1} \tag{A24}$$

Let us know what you think.

---

## Author Comment (AC4) · 4 Feb 2019

**In the following we use R2C1 (etc) to refer to comment 1 (C1) by reviewer 2 (R2).**

**Dr. S. Patidar**

R2C10: Dear Author

Thanks, You pointed it out right. I hope you must understand algebra and sign rules for operating the bracket. To cancel P(t) you need to enclose the third and fourth term within the brackets.

Response: Yes, we agree that it is better to understand the cancellation of P(t) by adding brackets around the third and fourth terms. Thanks for your suggestion.